# Impact of managerial overconfidence on abnormal audit fee: From the perspective of balance mechanism of shareholders

Xing-Xing He[1]◉, De-Cong Xie[1]◉, Ze-Min Hu[2]*, Xing-Li Bao[1]‡, Lin Li[1]‡

**1** Business School of Guilin University of Electronic Technology, Guilin University of Electronic Technology, Guilin, Guangxi, China, **2** Guilin University of Aerospace Technology, Guilin, Guangxi, China

◉ These authors contributed equally to this work.
‡ These authors also contributed equally to this work.
* zeminhu0409@163.com

**Data Availability Statement:** All relevant data are within the manuscript and its Supporting Information files.

**Funding:** The author(s) received no specific funding for this work.

## Abstract

Overconfidence, as a psychological feature that is difficult to measure, means that managers are overconfident in their management ability, investment judgment ability and knowledge richness, thus overestimating their ability and making irrational behavior. Based on the sample of Chinese listed firms from 2014 to 2018, we measure managerial overconfidence in terms of age, gender, education, position and salary, and analyzed the relationship between overconfidence, abnormal audit fees, and the balance mechanism of shareholders. The research results show that there is a significant positive correlation between managerial overconfidence and abnormal audit fees, and the balance mechanism of shareholders can significantly inhibit the positive correlation between managerial overconfidence and abnormal audit fees. The research results of this paper are conducive to the supervision department to further improve the relevant supervision measures, improve the audit quality, and provide theoretical support for the more specific requirements of audit fee information disclosure.

## 1 Introduction

Burrell first proposed behavioral finance in 1951, which applied psychology to the field of corporate finance research, thereby explaining many reasons that traditional economics cannot explain. For example, Hambrick and Mason [1] proposed the Upper Echelons Theory (UE Theory) that managers cannot grasp all information in a complex business environment, so they conduct business planning based on limited rationality. The characteristics inherent to managers will influence the decision-making [1], and influence the development of corporate financial reports [2]. As a psychological characteristic of managers, overconfidence is formed under the long-term influence of the external environment. Overconfidence will affect the behavior of managers and inevitably affect the business decisions of enterprises. Current research focuses on the economic consequences of managerial overconfidence and forms different perspectives: positive and negative. For example, overconfident management will result

**Competing interests:** The authors have declared that no competing interests exist.

in superior decision-making and propose competitive strategies; or overconfident management will lead to overinvestment and irrational mergers and acquisitions [3–6]. However, there are few studies on the impact of managerial overconfidence on external stakeholders, such as the auditor's pricing of firms with overconfident managers, especially abnormal audit fee. The economic meaning of abnormal audit fee is an important issue in audit theory. On the one hand, higher abnormal audit fees will reduce the independence of auditors [7] and lead to a decline in audit quality [8–10]. On the other hand, more audit fees may indicate that the auditors have put in more effort, and the auditors are committed to reducing accounting misstatements, so accounting quality should be improved, that is, abnormal audit fees should be positively related to accounting quality [11].

In China's unique economic and institutional background, most listed companies have evolved from state-owned enterprises. In most modern firms in China, state-owned shares account for a relatively high proportion, thus forming a phenomenon of ownership concentration. Although the share split reform has improved this situation, the phenomenon of ownership concentration is still relatively common. With the emergence of equity concentration, large shareholders have taken various measures to encroach on the company's resources and plunder the legitimate rights and interests of minority shareholders. In China, due to deficiencies in laws and inadequate corporate governance, ownership concentration provides an opportunity for large shareholders to obtain private profit. The phenomenon of tunneling behavior of large shareholders is more common. In order to suppress the tunneling behavior, the balance mechanism of shareholders has been heavily introduced into corporate governance. The balance mechanism of shareholders is a model of equity arrangement, which means that several major shareholders internally restrain each other to achieve the purpose of supervising the largest shareholder. The balance mechanism of shareholders can not only effectively restrain major shareholders from infringing on the interests of listed companies, but also supervise the decision-making behavior of managers and reduce irrational behavior of management. This article starts from the supervisory role of power balance with shareholders on managers, and explores whether balance mechanism of shareholders can significantly affect the changes in abnormal audit fees caused by managers' overconfidence.

Although some scholars have begun to pay attention to the relationship between managerial overconfidence and audit fee [12, 13], such studies are still incomplete. And there is not much research on Chinese capital markets. At the same time, because managerial overconfidence is a psychological feature, it is difficult to measure and judge in actual cases, but empirical research based on large sample data analysis may be able to better analyze the relationship between the two.

Our main contributions are: First, according to the UE theory, analyzing the impact of managerial overconfidence on abnormal audit fees, and provide more empirical evidence for external stakeholders to judge the reliability of corporate financial information. Second, analyzing the impact of the shareholder balance mechanism will help improve corporate governance and reduce the motivation of management to conduct earnings management and purchase audit opinions. Therefore, based on the personal characteristics of managers, this article explores its impact on abnormal audit fee, and provides more evidence for improving the audit work of Chinese listed companies.

## 2 Literature review and theoretical analysis

### 2.1 Managerial overconfidence and abnormal audit fee

In recent years, based on the Upper Echelons Theory, many scholars have done a lot of research on the characteristics of managers affecting company decisions. The theory is based

on the limited rationality of human beings, and highlights the role of manages' characteristics (gender, age, educational background, etc.) on managerial cognitive models and the impact on company performance. Faccio et al. [14] found that firms run by female CEOs have lower leverage, less volatile earnings, and a higher chance of survival. Huang et al. [15] and Sun et al. [16] analyzed the relationship between the age of the CEO and the quality of financial reports, and found that the older the CEO is, the lower the possibility of fraud in financial reporting and the more financial information reliable. Li et al. [17] used Thai companies as a sample and found that CEO characteristics affect environmental information disclosure. The level of education and tenure of the CEO's financial expertise has encouraged companies to disclose more environmental information. Elkhatib et al. [18] analyzed the influence of the CEO's personal relationship on the company's M & A structure, and found that company decisions may be affected by the status of the CEO in the social class, and CEOs with high social status will receive private benefits from it.

The managerial overconfidence specifically refers to the overconfidence of managers in their management ability, investment judgment ability and knowledge richness, which means that their judgment always deviates psychologically from the actual situation and overestimates their ability. Overconfidence, as a psychological feature that is difficult to measure for management, also affects business operations. Galasso et al. [19] build up a career concern model where CEOs innovate to provide evidence of their ability. Galasso believes that overconfident CEOs will underestimate the possibility of failure, and they are more likely to pursue innovation. It was found that management's overconfidence had a positive correlation with the number of enterprises' weighted patents. Hirshleifer et al. [20] found that companies with overconfident CEOs invest more in innovation, obtain more patents and patent citations, and achieve greater innovation success. Chen [21] found that the overconfidence of the CEO can be used as an explanation of the company's cash policy. In innovative companies, R&D activities require high and ongoing funding. In addition, overconfident CEOs are better equipped to meet these difficult challenges than non-overconfident CEOs. The results show that overconfidence is the key to motivating CEOs to hold cash. Campbell [22] found that managerial overconfidence will motivate them to work hard to achieve the goal of maximizing shareholder wealth. Chen [23] believes that over-confident managers are more likely to overestimate future demand, so when sales decline, the possibility of reducing (sales, general and administrative expenses) costs is smaller.

Larwood and Whittaker [24] found that overconfident individuals tend to exaggerate their skill level and believe that their level is higher than the average level of people. Langer [25] put forward the theory of control illusion, that over-confident individuals have an unreasonable expectation of the probability of successfully doing a thing higher than the objective probability of success of the matter. When there is "control illusion", people believe that their success rate is high and are more willing to take risks. Therefore, overconfidence will lead management to believe too much in their management capabilities, thereby making unreasonable decisions, overestimating the benefits of their investment projects and underestimating project risks [26]. For projects with a negative net present value, management believes that they can create more value through their own efforts. As a result, negative NPV projects will be retained for too long, and poor performance will accumulate, which may cause the stock price to plummet [27]. In addition, overconfident management overestimated their ability to generate revenue and were more willing to implement corporate mergers and acquisitions, which ultimately led to overpayment by the target company and reduced the value of the merger [28]. In order to disclose good profit forecasts and financial performance, management is more motivated to implement earnings management and is more willing to manipulate accruals [29]. Subramanyam [30] pointed out that the higher the degree of corporate earnings

management, the greater the litigation risk that certified public accountants need to bear, so they will raise audit fees to compensate for the risks.

Audit fee can be divided into normal audit fee and abnormal audit fee. The normal audit fee reflects the auditor's work cost and potential risks, and is composed of factors such as the client's asset size, business risk, and complexity [31]. Abnormal audit fee reflects the special economic contractual relationship between the client and the auditor and are determined by the client's special needs [10]. Unlike Europe and the United States and other countries that have strict and perfect supervision mechanisms for corporate audits, the external supervision of Chinese accounting firms and their audit work is still relatively weak. At this stage, there is still room for "rent-seeking" in the audit work of Chinese listed firms. On the one hand, higher levels of abnormal audit fees will increase the dependence of auditors on customers [32], leading to a decrease in the independence of auditors. Second, due to oversupply and competitive incentives in the Chinese audit market, auditors have no advantage in negotiations with management. In order to maintain the relationship with customers and continue to earn excess profits, auditors are more likely to condone the earnings management behavior of the management of listed firms [33]. On the other hand, the rewards and risks of indulging earnings management actions are not consistent. Signed certified public accountants enjoy the additional benefit of abnormal audit fee, and the harm caused by the exposure of the collusion event is shared by all auditors of the same accounting firm [34]. Therefore, as abnormal audit fee increase, corporate earnings management behavior is more likely to occur.

In addition, the possibility of the auditor's audit opinion on the company's first continuous operating revision is positively related to management's overconfidence [35]. Listed firms can purchase audit opinions through abnormal audit fees [36], while reducing audit quality [34]. In addition, it is difficult to observe the use of abnormal audit fee to purchase audit opinions. From the perspective of improving audit opinion, overconfident management are more likely to pay abnormal audit fees to reduce the level of auditors' disclosure of corporate earnings management and make the firms' operating performance better. We suggest:

Hypothesis $H1_1$: The relationship between managerial overconfidence and abnormal audit fee is positive.

Hypothesis $H1_0$: The relationship between managerial overconfidence and abnormal audit fees is either zero or negative.

## 2.2 Moderating role of balance mechanism of shareholders

Ownership concentration is an effective corporate governance mechanism [37]. During economic transition, equity concentration will promote higher profitability and labor productivity of the company [38]. However, Chinese small and medium-sized investor protection and shareholder behavior monitoring systems are still incomplete. The controlling shareholder mainly obtains private profits through the pyramid structure and participation in management, which seriously harms the interests of listed companies and their small and medium shareholders [39]. Excessive control of large shareholders has promoted potential hollowing out and other moral hazard activities [40]. In order to suppress the "tunnel behavior" of major shareholders, balance mechanism of shareholders has been introduced in corporate governance.

The balance mechanism of shareholders are the product of the inadequacy of Chinese laws and regulatory systems. We believe that shareholders are more likely to implement more effective supervision of the largest shareholder and management in order to protect their own interests. The balance mechanism of shareholders means that control is shared by several

major shareholders. It is a model of equity arrangement through the major shareholders' mutual supervision so that the controlling shareholders cannot control the decision alone [41]. It can suppress the tunnel behavior of the major shareholders through the role of supervision and avoid the deprivation of the interests of the minority shareholders by the controlling shareholders [42]. It also has a certain supervisory role on the managers. First, the balance mechanism of shareholders makes a significant impact on management appointments and management incentives, which is conducive to improving management incentives and supervision systems and reducing management inappropriate behavior [41]. Second, balanced shareholders can curb individual behavior by participating in corporate governance. When there is a more equal distribution of voting among major shareholders, no major shareholder can independently control the production operation and decision-making of the entire company [43], and company managers are no exception. And it can prevent the major shareholders and management from forming a conspiracy to invade the interests of small and medium shareholders. Third, the diversification of major shareholders can form an effective supervision of managers. In order to protect their own interests, shareholders have the incentive to monitor the behavior of major shareholders and managers, thereby forming more effective supervision. Finally, the quality of internal control of listed companies has been improved, reducing the overconfident managers' earnings management motivation and audit opinion purchase motivation. The positive impact of managerial overconfidence on abnormal audit fee has been weakened. Therefore, we recommend:

Hypothesis $H2_1$: The balance mechanism of shareholders has a negative adjustment to the relationship between overconfidence and abnormal audit fees.

Hypothesis $H2_0$: The balance mechanism of shareholders has a positive adjustment or a non-adjustment to the relationship between overconfidence and abnormal audit fees.

## 3 Research design

### 3.1 Sample and setting

We select listed manufacturing companies from 2014–2018 as the research sample. In order to avoid the impact of abnormal financial conditions on research, we removed companies with warnings of delisting risks. Considering that Initial Public Offering has an abnormal impact on the company's financial position, we exclude firms listed after 2014. Finally, a data set of 7145 firm-year observations from 2014 to 2018 is finally assembled for analysis. All research data were obtained from the CSMAR database. It is one of the reliable databases of Chinese listed company information, which can obtain all kinds of financial data [44]. And the data were processed and analyzed by SPSS25.0 and Eviews 9.0 software (S1 Table).

### 3.2 Measures

In related literature, the main measurement methods of overconfidence (*OVERC*) mainly include: executive option implementation [3, 4]; external mainstream media evaluation method [29, 45] and performance forecast deviation [29]. The external mainstream media evaluation method is widely used by international scholars. However, due to the lack of a relevant database in China and the firms' managers to maintain a close relationship with the media, the external mainstream media assessment method is not feasible. At the same time, the time gap between the release of performance forecasts and performance reports by Chinese companies is relatively short. Management generally knows the actual performance and makes

accurate performance forecasts. Therefore, the performance forecast deviation method is not in line with China's actual situation. Considering the availability and feasibility of the data, we use the background characteristics of the CEO to measure whether the management is overconfident.

Overconfidence is a psychological deviation caused by individual characteristics, such as the CEO's gender, age, education, position, and salary. In business management, men are more radical and conceited than women [46]. Older managers are more inclined to avoid risks and act cautiously. People with higher education levels are more confident in their own abilities and judgments, and are more likely to show overconfidence [47]. The CEO with the chairmanship will recognize his own abilities more and show a tendency to overconfidence in decision-making [47]. The high salary that the company pays to the CEO will bring him positive psychological feedback, which will enable the CEO to show stronger self-esteem and generate overconfidence [48]. Therefore, we set gender indicator, age indicator, education indicator, position indicator and salary indicator. If the CEO is male, the gender indicator is assigned a value of 1, otherwise it is 0; if the CEO's age is less than the sample median, the age indicator is assigned a value of 1, otherwise it is 0; if the CEO has a bachelor's degree or above, the educational indicator is assigned a value of 1, otherwise 0; If the CEO has the position of chairman, the position indicator is assigned a value of 1, otherwise it is 0; if the ratio of the sum of the top three executives' salaries to the sum of all executives' salaries is greater than the sample average, the salary indicator is assigned a value of 1, otherwise it is 0. We set up dummy variables (*OVERC*). If the sum of the above indicators of the sample is not less than 4, it will be considered overconfident, and *OVERC* is assigned a value of 1, otherwise it is 0.

Abnormal audit fee (*AAF*) is the variable to be explained. Based on the Simunic model [49] and related extended literature [7–11], we establish the following audit fee model. The explanatory variables of this model are the influencing factor of audit charges, and its fitted value is the normal charge that can be explained. The difference between the actual audit fee and the normal fee, which is the residual item, is an abnormal audit fee that cannot be explained by the influencing factors.

$$FEE_{i,t} = \alpha_0 + \alpha_1 SIZE_{i,t} + \alpha_2 INV_{i,t} + \alpha_3 REC_{i,t} + \alpha_4 ROA_{i,t} + \alpha_5 LEV_{i,t} +$$
$$\alpha_6 LOSS_{i,t} + \alpha_7 PRENU_{i,t} + \alpha_8 BIG4_{i,t} + \alpha_9 EMP_{i,t} + \sum YEAR + \sum IND + \varepsilon_{i,t} \tag{1}$$

The above model (1) considers many factors that affect audit fees. *FEE* represents the natural logarithm of actual audit fee. *SIZE* represents the size of the firms. The audit of large-scale firm is more complicated and the workload is larger. *INV* represents the proportion of the firms' inventory. *REC* represents the proportion of the firms' accounts receivable. The higher *INV* and *REC*, the more audit procedures and the greater the audit workload. *ROA* represents the profitability of an enterprise. *LEV* represents the firms' debt situation. *LOSS* represents the financial situation. Both the debt situation and the financial situation will affect the audit risk. *PRENU* represents the audit opinion of the previous year. *BIG4* represents the scale of accounting firms, and the fees of the Big Four accounting firms are higher. *EMP* is the number of employees in an enterprise and represents the complexity of the firms' business. The more complex the business, the more audit procedures and the greater the audit workload. *YEAR* represents a dummy variable of time. *IND* stands for industry dummy variable. The residual term $\varepsilon$ represents the abnormal audit fees.

Balance mechanism of shareholders (*EQUITY*) means that several major shareholders share control. Through the mutual control between shareholders, any major shareholder cannot control the decision of the enterprise individually, thus realizing the mutual supervision equity arrangement model. We use the ratio of the sum of the shareholding ratios of the

second to tenth largest shareholders to the shareholding ratio of the first largest shareholder. The larger the indicator, the smaller the degree of control of the largest shareholder, and the greater the role of supervision.

We control for several factors that may affect abnormal audit fees: firms' size (*SIZE*), the firm's debt situation (*LEV*), profitability (*ROA*), audit opinion (*OP*), firm loss (*LOSS*), accounting firms (*BIG4*), business complexity (*EMP*), inventory ratio(*INV*), proportion of accounts receivable (*REC*), audit opinion of the previous year(*PRENU*), proportion of independent directors(*ID*), board of supervisors(*BS*).

## 3.3 Methodology

Model (2) is used to examine the impact of overconfident on abnormal audit fee. Model (3) is used to examine the moderating effect of balance mechanism of shareholders.

$$AAF_{i,t} = \beta_0 + \beta_1 OVERC_{i,t} + \sum \beta_k Controls_{i,t} + \sum YEAR + \sum IND + \varepsilon_{i,t} \qquad (2)$$

$$\begin{aligned} AAF_{i,t} = \\ \gamma_0 + \gamma_1 OVERC_{i,t} + \gamma_2 EQUITY_{i,t} + \gamma_3 OVERC_{i,t} \times EQUITY_{i,t} + \sum \gamma_k Controls_{i,t} + \\ \sum YEAR + \sum IND + \varepsilon_{i,t} \end{aligned} \qquad (3)$$

Where *OVERC×EQUITY* is defined as moderating effect of balance mechanism of shareholders. If Hypothesis H1$_1$ is true, β$_1$ should be positive and significant, and if Hypothesis H2$_1$ is true, γ$_3$ should be negative and significant. Table 1 shows all the variables and their definitions.

**Table 1. Variable definition table.**

| Variable type | Variable code | Variable definitions |
|---|---|---|
| Explained variable | *AAF* | The difference between the actual audit fees and the normal fees |
| Explanatory variables | *OVERC* | The dummy variable. = 1 if the sum of the five indicators is not less than 4 and 0 otherwise |
| Moderator | *EQUITY* | The sum of the shareholding ratios of the second to tenth largest shareholders / the ratio of the first largest shareholder. |
| Control variable | *SIZE* | Natural logarithm of total assets |
| | *INV* | Net inventory / total assets |
| | *REC* | Net accounts receivable / total assets |
| | *ROA* | Net profit / total assets |
| | *LEV* | Debt / total assets |
| | *LOSS* | The dummy variable. = 1 if net profit for the year is below zero and 0 otherwise |
| | *PRENU* | The dummy variable. = 1 if the non-standard opinion is issued previous year and 0 otherwise |
| | *BIG4* | The dummy variable. = 1 if accounting firm is Big Four and 0 otherwise |
| | *EMP* | Square root of the number of employees |
| | *ID* | Number of independent directors / board of directors |
| | *BS* | Natural logarithm of the number of supervisors |
| | *OP* | The dummy variable. = 1 if the non-standard opinion is issued and 0 otherwise |
| | *YEAR* | Time dummy variable |
| | *IND* | Industry dummy variables |

**Table 2. Descriptive statistics.**

| Variable | Mean | SD | MIN. | MAX. | Observed |
|---|---|---|---|---|---|
| AAF | 0.007 | 0.471 | -2.257 | 3.221 | 7145 |
| OVERC | 0.310 | 0.462 | 0.000 | 1.000 | 7145 |
| EQUITY | 0.984 | 0.824 | 0.005 | 7.191 | 7145 |
| SIZE | 22.320 | 1.246 | 18.287 | 28.253 | 7145 |
| INV | 0.140 | 0.134 | 0.000 | 0.922 | 7145 |
| REC | 0.128 | 0.110 | 0.000 | 0.810 | 7145 |
| ROA | 0.041 | 0.071 | -1.859 | 0.964 | 7145 |
| LEV | 0.416 | 0.203 | 0.009 | 2.394 | 7145 |
| LOSS | 0.090 | 0.281 | 0.000 | 1.000 | 7145 |
| PRENU | 0.020 | 0.138 | 0.000 | 1.000 | 7145 |
| BIG4 | 0.060 | 0.230 | 0.000 | 1.000 | 7145 |
| EMP | 60.714 | 50.701 | 0.000 | 672.020 | 7145 |
| ID | 0.376 | 0.057 | 0.231 | 0.800 | 7145 |
| BS | 1.220 | 0.244 | 0.000 | 2.485 | 7145 |
| OP | 0.020 | 0.145 | 0.000 | 1.000 | 7145 |

## 4 Empirical analysis

### 4.1 Descriptive statistics

The descriptive statistics on variables introduced to our model is shown in Table 2. In a sample of 1429 firms in China from 2014 to 2018, the average value of *AAF* is 0.007. This means that the audit work of Chinese listed firms has excessive charging problems. The average value of *OVERC* is 0.310, showing that the level of managerial overconfidence in Chinese firms is low. The average value of *EQUITY* is 0.984, indicating that the balance mechanism of shareholders can play a supervisory role to a certain extent. The average value of *INV* is 0.140, and the average value of *REC* is 0.128, indicating that Chinese firms have low accounts receivable and inventory occupancy. The average *LEV* is 0.416, indicating that the enterprise has less debt. The average value of *LOSS* is 0.090, indicating that most companies have achieved profitability. The average value of *PRENU* is 0.020 and the average value of *OP* is 0.020, indicating that few companies have been issued with non-standard audit opinions.

### 4.2 Correlation analysis

The correlations matrix of this study is shown in Table 3. Abnormal audit fees and managerial overconfidence are significantly correlated at the 1% level, and their correlation coefficient is 0.095. It is preliminarily verified that managerial overconfidence has a positive and significant impact on abnormal audit fees. By observing the data distribution, we believe that the reason for the smaller correlation coefficient is the different range of variables. The range of *AAF* is (-2.2, 3.2), and average value is 0.007; the range of *OVERC* is (0,1), and average value is 0.310. Balance mechanism of shareholders and abnormal audit fees are significantly correlated at the 1% level. In addition, all correlation coefficients are less than 0.3, and there is no severe collinearity problem between the variables.

### 4.3 Regression analysis

We use SPSS 25.0 software to analyze data. For Hypothesis H1$_1$, we propose that managerial overconfidence will raise abnormal audit fee for firms. Model (2) in Table 4 show that effect of overconfidence on abnormal audit fee are significant and positive ($\beta$ = 0.042, p < 0.01),

**Table 3. Correlation coefficient table.**

|  | *AAF* | *OVERC* | *EQUITY* | *SIZE* | *INV* | *REC* | *ROA* | *LEV* | *LOSS* | *PRENU* | *BIG4* | *EMP* | *ID* | *BS* | *OP* |
|---|---|---|---|---|---|---|---|---|---|---|---|---|---|---|---|
| *AAF* | 1.000 |  |  |  |  |  |  |  |  |  |  |  |  |  |  |
| *OVERC* | 0.095** | 1.000 |  |  |  |  |  |  |  |  |  |  |  |  |  |
| *EQUITY* | 0.075** | 0.055** | 1.000 |  |  |  |  |  |  |  |  |  |  |  |  |
| *SIZE* | -0.280** | -0.164** | -0.066** | 1.000 |  |  |  |  |  |  |  |  |  |  |  |
| *INV* | -0.016 | -0.028* | -0.079** | 0.151** | 1 |  |  |  |  |  |  |  |  |  |  |
| *REC* | -0.018 | 0.015 | 0.087** | -0.141** | -0.099** | 1 |  |  |  |  |  |  |  |  |  |
| *ROA* | -0.009 | 0.026* | -0.004 | 0.042** | -0.091** | 0.008 | 1 |  |  |  |  |  |  |  |  |
| *LEV* | -0.032** | -0.088** | -0.110** | 0.520** | 0.329** | 0.043** | -0.292** | 1 |  |  |  |  |  |  |  |
| *LOSS* | -0.005 | 0.014 | -0.005 | -0.088** | 0.000 | -0.023* | -0.555** | 0.134** | 1 |  |  |  |  |  |  |
| *PRENU* | -0.002 | 0.003 | 0.017 | -0.059** | 0.005 | -0.031** | -0.097** | 0.099** | 0.102** | 1 |  |  |  |  |  |
| *BIG4* | -0.004 | -0.050** | -0.035** | 0.412** | -0.021 | -0.087** | 0.041** | 0.135** | -0.036** | -0.021 | 1 |  |  |  |  |
| *EMP* | -0.019 | -0.122** | -0.055** | 0.718** | 0.006 | -0.061** | 0.047** | 0.311** | -0.050** | -0.034** | 0.442** | 1 |  |  |  |
| *ID* | 0.036** | 0.108** | -0.024* | 0.003 | 0.012 | 0.016 | -0.035** | -0.017 | 0.039** | -0.014 | 0.02 | 0.061** | 1 |  |  |
| *BS* | -0.090** | -0.160** | -0.103** | 0.267** | 0.002 | -0.130** | -0.029* | 0.199** | 0.008 | 0.003 | 0.124** | 0.223** | -0.118** | 1 |  |
| *OP* | 0.039** | 0.007 | 0.033** | -0.054** | 0.000 | -0.030* | -0.226** | 0.107** | 0.177** | 0.455** | -0.028* | -0.041** | -0.003 | -0.011 | 1 |

**, and * indicate significance at the 5% and 10% levels, respectively.

**Table 4. Results of regression analysis.**

| Variable name | *AAF* | |
|---|---|---|
|  | **Model (2)** | **Model (3)** |
| *OVERC* | 0.042*** (3.916) | 0.062*** (3.677) |
| *EQUITY* |  | 0.089*** (7.027) |
| *OVERC × EQUITY* |  | -0.033* (-1.787) |
| *SIZE* | -0.821*** (-43.929) | -0.824*** (-44.181) |
| *INV* | 0.016 (1.411) | 0.019* (1.686) |
| *REC* | -0.105*** (-9.606) | -0.112*** (-10.265) |
| *ROA* | 0.059*** (4.414) | 0.062*** (4.693) |
| *LEV* | 0.26*** (18.146) | 0.268*** (18.754) |
| *LOSS* | -0.059*** (-4.695) | -0.058*** (-4.625) |
| *PRENU* | -0.065*** (-5.514) | -0.066*** (-5.627) |
| *BIG4* | 0.09*** (7.609) | 0.09*** (7.625) |
| *EMP* | 0.438*** (27.576) | 0.44*** (27.806) |
| *ID* | 0.004 (0.351) | 0.007 (0.616) |
| *BS* | -0.041*** (-3.689) | -0.035*** (-3.121) |
| *OP* | 0.031** (2.586) | 0.028** (2.373) |
| *Industry* | Included | Included |
| *Year* | Included | Included |
| *Constant* | 6.705*** | 6.642*** |
| *F value* | 117.675*** | 109.507*** |
| *Adj.R²* | 0.178 | 0.233 |

***, **, and * indicate significance at the 1%, 5%, and 10% levels, respectively. T value is in parentheses.

revealing that the null hypothesis of no relationship or a negative relationship between *OVERC* and *AAF* could be rejected at the 1% probability level. Therefore, there is a reliable positive correlation between managerial overconfidence and abnormal audit fee, and hypothesis $H1_1$ is established. Additionally, in the absence of control variables, the partial correlation coefficient between *AAF* and *OVERC* is 0.048, which is significant at the 0.05 level. In the presence of control variables, the partial correlation coefficient between *AAF* and *OVERC* is 0.034, which is significant at the 0.05 level. The adjusted R-square for the regression equation was 0.178 and most of the control variables were significant at the 1% level which indicates the equation was reliable.

Hypothesis $H2_1$ predicts that balance mechanism of shareholders will weaken the positive relationship between overconfidence and abnormal audit fee. Model (3) in Table 4 shows that the interaction between managerial overconfidence and balance mechanism of shareholders has a significant negative relationship with abnormal audit fee ($\gamma = -0.033$, $0.05 < p < 0.1$), revealing that the null hypothesis of positive adjustment or non-adjustment of balance mechanism of shareholders is rejected at a probability level of 0.1. The moderate effect of balance mechanism of shareholders is useful in Chinese listed firms. Hypothesis $H2_1$ is established. The R-square of 0.233 and the fact that most of the control variables were significant at the 0.01 level indicates the results of the analysis were dependable.

## 5 Conclusion

We used Chinese listed companies from 2014 to 2018 as a sample and empirically tested the impact of managerial overconfidence on abnormal audit fees. At the same time, combined with the perspective of balance mechanism of shareholders, the role of small and medium-sized shareholders in monitoring checks and balances is studied. It was found that managerial overconfidence was significantly positively correlated with abnormal audit fees, and the balance mechanism of shareholders could significantly inhibit the positive relationship between management overconfidence and abnormal audit costs. Research reveals that overconfident management overestimates their own abilities and deviations in the accuracy of judgments, which can easily put companies in financial trouble. In order to disclose good profit forecasts and financial performance, overconfident management is more motivated to implement earnings management and reduce the level of auditors' disclosure of corporate earnings management through abnormal audit fees, making the company's operating performance more perfect. The balance mechanism of shareholders can effectively realize the role of corporate governance, reduce the irrational behavior of management, and reduce the motivation for earnings management and the motivation to purchase audit opinions. Therefore, it is necessary for the regulatory department to further improve the relevant disciplinary measures, strengthen the monitoring of abnormal audit fee, and require the disclosure of more specific audit fee information. Finally, the impact of purchasing audit opinions or earnings management on audit quality is avoided. It is necessary to accelerate the improvement of the equity structure of listed companies, and deeply integrate equity checks and balances with corporate governance. This allows diversified small and medium shareholders to give full play to their supervision and governance to protect the legitimate rights and interests of stakeholders.

The Chinese market has some distinctive characteristics. For example, the state-owned ownership of the enterprise is large, and the monarch-subject thought spread for thousands of years in China. These characteristics are conducive to the formation of overconfidence in management, leading management to make decisions that deviate from reality and are reflected in abnormal audit fees. In recent years, the China Securities Regulatory Commission has increased the investigation and punishment of illegal activities of listed companies and

accounting firms, and announced administrative penalty decisions, including the conspiracy of management and auditors. This further reflects the flaws in China's regulatory system. This study analyzes the supervisory role of balance mechanism of shareholders from the perspective of equity structure. In future research, we can further explore the supervisory role of the nature of equity, such as state-owned shares and institutional investors.

In addition, abnormal audit fee can be divided into positive and negative, reflecting the different contractual relationships between management and auditors. This study discusses overconfident management paying excessive audit fee to obtain favorable conditions. Future research can be extended to the bargaining power of overconfident management, that is, the discounts of fee that are held in economic negotiations, forming negative abnormal audit fee. Whether this has an impact on the firms' audit quality, audit opinions and accounting information transparency is worth discussing.

## Supporting information

**S1 Table. The raw data of this paper.**
(XLS)

## Author Contributions

**Conceptualization:** De-Cong Xie, Xing-Li Bao, Lin Li.

**Data curation:** De-Cong Xie, Xing-Li Bao, Lin Li.

**Methodology:** Xing-Li Bao.

**Resources:** De-Cong Xie, Ze-Min Hu, Lin Li.

**Supervision:** Ze-Min Hu.

**Writing – original draft:** Xing-Xing He.

**Writing – review & editing:** Xing-Xing He.

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
