## [Decision Letter · Decision Letter 0]

24 Jun 2020

PONE-D-20-10813

Impact of Managerial Overconfidence on Abnormal Audit Fee: from the Perspective of Balance Mechanism of Shareholders

PLOS ONE

Dear Dr. Hu,

Thank you for submitting your manuscript to PLOS ONE. After careful consideration, we feel that it has merit but does not fully meet PLOS ONE’s publication criteria as it currently stands. Therefore, we invite you to submit a revised version of the manuscript that addresses the points raised during the review process.

We look forward to receiving your revised manuscript.

Kind regards,

Ke-Chiun Chang

Academic Editor

PLOS ONE

Journal Requirements:

3. We note you have included a table to which you do not refer in the text of your manuscript. Please ensure that you refer to Table 1 in your text; if accepted, production will need this reference to link the reader to the Table.

Reviewers' comments:

Reviewer's Responses to Questions

**Comments to the Author**

1. Is the manuscript technically sound, and do the data support the conclusions?

Reviewer #1: Yes

Reviewer #2: Yes

Reviewer #3: Partly

2. Has the statistical analysis been performed appropriately and rigorously? 

Reviewer #1: Yes

Reviewer #2: Yes

Reviewer #3: No

3. Have the authors made all data underlying the findings in their manuscript fully available?

Reviewer #1: Yes

Reviewer #2: Yes

Reviewer #3: Yes

4. Is the manuscript presented in an intelligible fashion and written in standard English?

Reviewer #1: Yes

Reviewer #2: Yes

Reviewer #3: Yes

5. Review Comments to the Author

Reviewer #1: 1.The data is sorted and processed to support its conclusions reasonably.

2.The use of quantitative methods can be directed to the subject of the article.

3.The authors provides complete data in their manuscript fully available.

4.The manuscript is presented in an intelligible fashion and written in standard English.

5.The impact of management overconfidence on audit fees in China is an interesting topic.

6.The author can add some future research directions in China to the article.

Reviewer #2: The structure of the article is reasonable and can be expressed in English. All the data can be obtained, and it can also provide strong support for the conclusion of the article. The quantitative analysis method used can reasonably target the article theme.

The subject of the article is the impact of management overconfidence on audit costs in China. This is a rare article type and quite interesting. In China, the influence of government policies is stronger than market forces. Therefore, policies can effectively change the market conditions, which also shows the differences between China and the West.

The recommendations can be analyzed by different industries, and there may be interesting results.

Reviewer #3: Using a convenience sample and some simple linear models the investigators have concluded that the research results show that there is a positive correlation between management overconfidence and abnormal audit fee, and the shareholder's equilibrium mechanism can inhibit the positive correlation between management overconfidence and abnormal audit costs. There are several issues. Specifically:

1. The investigators are using linear models throughout. Such is not always the case for economic models. There should be some goodness of fit considerations in this development.

2. The correlations of Table 3 look rather weak, although some are significant. Some discussion is needed in this case.

3. The results of the regression analysis is seen in Table 4. Again, the beta values for OVERC are less than 0.10. What is the partial or semi partial correlations of this value with the dependent measure?

4. Hypotheses H1 and H2 should be expressed as null hypotheses in a true significance test.

6. PLOS authors have the option to publish the peer review history of their article (what does this mean?). If published, this will include your full peer review and any attached files.

Reviewer #1: No

Reviewer #2: No

Reviewer #3: No

---

## [Author Response · Author response to Decision Letter 0]

7 Aug 2020

Dear Editor / Reviewer: 

First of all, we sincerely thank the responsible editors and reviewers of “PLOS ONE” for your review of the format of this paper in your busy work and provide constructive and valuable suggestions for this paper. We revised the manuscript in accordance with the reviewers’ comments, and carefully proof-read the manuscript to minimize typographical, grammatical, and bibliographical errors. Here below is our description on revision according to the reviewers’ comments.

【Editor’s Comment 1】:

【Authors’ Response】:

The layout and format guidelines have been followed.

【Editor’s Comment 2】:

Please ensure that you have an ORCID iD and that it is validated in Editorial Manager.

【Authors’ Response】:

The corresponding author already has ORCID iD and has been verified in the Editorial Manager.

【Editor’s Comment 3】:

Please ensure that you refer to Table 1 in your text; if accepted, production will need this reference to link the reader to the Table.

【Authors’ Response】:

Table 1 has been referred in the text and the description of Table 1 has been made on page 18.

【Editor’s Comment 4】:

Please include captions for your Supporting Information files at the end of your manuscript, and update any in-text citations to match accordingly.

【Authors’ Response】:

The captions of the Support Information files have been appended at the end of the manuscript, and any in-text citations have been updated.

【Reviewer 1’s Comment 1】: 

The author can add some future research directions in China to the article.

【Authors’ Response】:

Thank you for reading our manuscript and your recognition of our research. We will continue our efforts in academic research to make more achievements. The manuscript has been revised based on your suggestions and comments. We have added the following discussion to further explain some future research directions in China on page 27.

The Chinese market has some distinctive characteristics. For example, the state-owned ownership of the enterprise is large, and the monarch-subject thought spread for thousands of years in China. These characteristics are conducive to the formation of overconfidence in management, leading management to make decisions that deviate from reality and are reflected in abnormal audit fees. In recent years, the China Securities Regulatory Commission has increased the investigation and punishment of illegal activities of listed companies and accounting firms, and announced a number of administrative penalty decisions, including the conspiracy of management and auditors. It further reflects the flaws in China's regulatory system. This study analyzes the supervisory role of balance mechanism of shareholders from the perspective of equity structure. In future research, we can further explore the supervisory role of the nature of equity, such as state-owned shares and institutional investors.

In addition, abnormal audit fees can be divided into positive and negative, reflecting the different contractual relationships between management and auditors. This study discusses overconfident management paying excessive audit fees to obtain favorable conditions. Future research can be extended to the bargaining power of overconfident management, that is, the discounts of fees that are held in economic negotiations, forming negative abnormal audit fees. Its impact on corporate audit quality, audit opinion, and transparency of accounting information is unknown and worth looking forward to.

【Reviewer 2’s Comment 1】: 

The subject of the article is the impact of management overconfidence on audit costs in China. This is a rare article type and quite interesting. In China, the influence of government policies is stronger than market forces. Therefore, policies can effectively change the market conditions, which also shows the differences between China and the West.

【Authors’ Response】:

Thank you for reading our manuscript and your recognition of our research. We will continue our efforts in academic research to make more achievements. China is currently in a stage of transition, which provides us with more new perspectives and samples for research. The characteristics of the Chinese market are different from those of Western countries, and the impact and mechanism of managerial overconfidence may be different. For example, the state-owned ownership of enterprises is large, and the monarch-subject thought spread for thousands of years in China. These characteristics are conducive to the formation of overconfidence in management. In China, the impact of management overconfidence may be more profound than in Western countries. Therefore, our research provides more evidence for the field of overconfidence.

【Reviewer 2’s Comment 2】: 

The recommendations can be analyzed by different industries, and there may be interesting results.

【Authors’ Response】:

Thank you for your profound analysis and practical advice. Our current research focuses on the analysis of Chinese companies as a whole, and intends to get a conclusion that can be widely used. After drawing the conclusion that management overconfidence will increase abnormal audit fee, we will conduct more in-depth and innovative research. Thank you very much for your industry factors. We discuss that industry factors do have an important impact on the relationship between management overconfidence and abnormal audit costs. Your suggestion is very important to us. Our next research will take into account the characteristics of different industries and conduct more detailed research.

【Reviewer 3’s Comment 1】:

The investigators are using linear models throughout. Such is not always the case for economic models. There should be some goodness of fit considerations in this development.

【Authors’ Response】:

Through literature review and theoretical analysis, we found that the psychological characteristics of overconfidence in management have an impact on corporate decision-making. For example, overconfidence will: (a) trigger irrational investment behavior and damage investment value; (b) invest more in innovation and obtain innovative results; (c) work harder and so on.

Overconfident people think that they have superior management skills and have unreasonable expectations about the possibility of accomplishing something successfully. Therefore, overconfident management believes that they have a high chance of success and are more willing to take risks. In terms of investment projects, over-confident management will overestimate returns and underestimate risks, which ultimately leads to the failure of investment projects. In terms of corporate mergers and acquisitions, overconfident management is more willing to implement mergers and acquisitions and pay more. Overconfident management is more motivated to implement earnings management to cover up their failures in business operations.

The supervision of audit work in China is relatively weak, and the management can reach a special secret contractual relationship with the auditor through abnormal audit fees. For overconfident management, the auditor's dependence on customers can be increased through higher levels of abnormal audit fees, which leads to a decline in the auditor's independence. The types of audit opinions and the content of audit reports will also be interfered with by the will of management. In addition, the use of abnormal audit fees to purchase audit opinions is secretive and difficult to detect. From the perspective of improving audit opinions, overconfident management is more likely to pay abnormal audit fees in order to reduce the auditor's level of disclosure of the company's earnings management, and make the company's operating performance better.

As far as auditors are concerned, due to the oversupply in the Chinese audit market, they have no advantage in negotiations with management. In order to maintain relationships with customers and continue to earn excess profits, auditors are more likely to condone the earnings management behavior of listed company management. The benefits and risks of laissez-faire earnings management are inconsistent. The signed CPA has the exclusive benefit, while the risk is shared by all auditors.

Through balance mechanism of shareholders, management incentives and supervision systems have been improved, and management misconduct has been reduced. Balanced shareholders can suppress individual behavior by participating in corporate governance. The diversification of major shareholders can form effective supervision of managers. Ultimately, the balance mechanism of shareholders has improved the quality of internal control of listed firms and reduced the earnings management motivation of overconfident managers.

Therefore, we believe that there is a reliable positive correlation between managerial overconfidence and abnormal audit fees, and the balance mechanism of shareholders can weaken this positive correlation. We learn from Malmendier (2011), Ahmed (2013), Duellman (2015), Mitra (2019), using linear equations to test the correlation between them. In the research design, we also considered more factors.

Regarding the balance mechanism of shareholders, in order to enhance reliability, we select the ratio of the sum of the shares of the second largest shareholder to the tenth largest shareholder to the share of the largest shareholder.

Regarding abnormal audit fees, we selected factors that affect the pricing of audit fees: scale, inventory, accounts receivable, the company's profitability, debt scale, financial status, audit opinions from the previous year, and number of employees. Through these factors, we get the normal audit fees and abnormal audit fees.

Regarding managerial overconfidence, we studied factors other than compensation and changed the OVERC measurement method. Due to changes in measurement methods, we deleted some firms with missing data from the original sample. We deleted 52 firms and got a data set containing 7145 observations. The overconfidence of management comes from personal characteristics and environmental factors (Hribar & Yang, 2016). According to the CEO's characteristics and his environment, his psychological characteristics can be distinguished. Personal characteristics such as gender, age, education, position, and salary level are sources of overconfidence. Therefore, we have established five indicators, gender indicator, age indicator, education indicator, position indicator and salary indicator. In business management, men are more radical and conceited than women. Older managers are more inclined to avoid risks and act cautiously. People with higher education have more confidence in their abilities and judgment, and are more likely to show overconfidence. The CEO who serves as the chairman will exaggerate his abilities more and show a tendency to become overconfident in decision-making. The high salary the company pays to the CEO will give him positive psychological feedback, which will enable the CEO to show stronger self-esteem and overconfidence. If the sample meets the following four or five conditions: male, young, highly educated, possesses both a CEO position and a chairman position, and a high salary level, it can be judged to be overconfidence. Finally, we conclude that there is a positive correlation between managerial overconfidence and abnormal audit fees, the balance mechanism of shareholders has a negative adjustment to the relationship between overconfidence and abnormal audit fees.

【Reviewer 3’s Comment 2】:

The correlations of Table 3 look rather weak, although some are significant. Some discussion is needed in this case.

【Authors’ Response】:

Based on literature reviews and theoretical analysis, we predict that there is a positive correlation between managerial overconfidence and abnormal audit fee. Through correlation analysis, we aim to determine the correlation between them and whether it is possible to conduct follow-up research. Table 3 on page 23 of the manuscript shows that the correlation coefficient between overconfidence (OVERC) and abnormal audit fee (AAF) is 0.095, which is significant at the 1% level. By observing the data distribution, we believe that the reason for the smaller correlation coefficient is the different range of variables. The range of AAF is (-2.2, 3.2), and average value is 0.007; the range of OVERC is (0,1), and average value is 0.310. Therefore, we preliminarily judge that there is a positive correlation between the managerial overconfidence and the abnormal audit fee. We are able to perform regression analysis.

【Reviewer 3’s Comment 3】:

The results of the regression analysis are seen in Table 4. Again, the beta values for OVERC are less than 0.10. What is the partial or semi partial correlations of this value with the dependent measure?

【Authors’ Response】:

Thank you for your valuable suggestions. We conducted a partial correlation analysis. As shown on page 24, in the absence of control variables, the correlation coefficient between OVERC and AAF is 0.048 (P<0.001). After adding the control variables, the correlation coefficient between OVERC and AAF is 0.034 (P<0.01). This shows that there is a reliable correlation between overconfidence and abnormal audit fees.

【Reviewer 3’s Comment 4】:

Hypotheses H1 and H2 should be expressed as null hypotheses in a true significance test.

【Authors’ Response】:

We revised the hypothesis on page 10 and page 13 of the manuscript. Hypothesis H1 and H2 have been expressed as the null hypothesis. Additionally, we added T values and F values to Table 4 on page 25 to test the null hypothesis. In model (2), we tested Hypothesis H11, the t value of OVERC was 3.916, and the coefficient was significant at the level of 1%, revealing that the null hypothesis of no relationship or a negative relationship between OVERC and AAF could be rejected at the 1% level. The F value for the equation is 117.675, which is significant at the level of 1%.

In model (3), we tested Hypothesis H21, the T value of OVERC×EQUITY is -1.787, and the coefficient is significant at the 10% level. The null hypothesis of positive adjustment or non-adjustment of balance mechanism of shareholders is rejected at a probability level of 0.1. The F value for the equation is 109.507, which is significant at the level of 1%.

We look forward to hearing from you regarding our submission. We would be glad to respond to any further questions and comments that you may have.

Thank you.

Reference

1. Duellman, S., Hurwitz, H., & Sun, Y. (2015). Managerial Overconfidence and Audit Fees. Journal of Contemporary Accounting & Economics, 11(2), 148-165.

2. Mitra, S., Jaggi, B., & Alhayale, T. (2019). Managerial overconfidence, ability, firm-governance and audit fees. Review of Quantitative Finance and Accounting, 52(3), 841-870.

3. Malmendier, U., Tate, G. A., & Yan, J. (2011). Overconfidence and Early-Life Experiences: The Effect of Managerial Traits on Corporate Financial Policies. Journal of Finance, 66(5), 1687-1733.

4. Ahmed, A. S., & Duellman, S. (2013). Managerial Overconfidence and Accounting Conservatism: managerial overconfidence. Journal of Accounting Research, 51(1), 1-30.

5. Fitriany, Veronica, S., & Anggraita, V. (2016). Impact of Abnormal Audit Fee to Audit Quality: Indonesian Case Study. American Journal of Economics, 6(1), 72-78.

6. Xie, z. Cai, C., & Ye, J. (2010). Abnormal Audit Fees and Audit Opinion - Further Evidence from China' s Capital Market.China Journal/ of Accounting Research, 3,51-70.

6. Hribar P, Yang H. Does CEO Overconfidence Affect Management Forecasting and Subsequent Earnings Management?. Contemporary Accounting Research.2015;33(1):204-227.

---

## [Editor Report · Decision Letter 1]

18 Aug 2020

Impact of managerial overconfidence on abnormal audit fee: From the perspective of balance mechanism of shareholders

PONE-D-20-10813R1

Dear Dr. Hu,

We’re pleased to inform you that your manuscript has been judged scientifically suitable for publication and will be formally accepted for publication once it meets all outstanding technical requirements.

Kind regards,

Ke-Chiun Chang

Academic Editor

PLOS ONE

---

## [Editor Report · Acceptance letter]

28 Aug 2020

PONE-D-20-10813R1 

Impact of managerial overconfidence on abnormal audit fee: From the perspective of balance mechanism of shareholders 

Dear Dr. Hu:

I'm pleased to inform you that your manuscript has been deemed suitable for publication in PLOS ONE. Congratulations! Your manuscript is now with our production department. 

Kind regards, 

on behalf of

Dr. Ke-Chiun Chang 

Academic Editor

PLOS ONE